# Epigenetic and Transcriptional Control of the Opioid Prodynorphine Gene: In-Depth Analysis in the Human Brain

**DOI:** 10.3390/molecules26113458

**Published:** 2021-06-07

**Authors:** Olga Nosova, Igor Bazov, Victor Karpyak, Mathias Hallberg, Georgy Bakalkin

**Affiliations:** 1Department of Pharmaceutical Biosciences, Uppsala University, 75124 Uppsala, Sweden; Igor.bazov@oru.se (I.B.); Mathias.Hallberg@farmbio.uu.se (M.H.); 2Mayo Clinic, Rochester, MN 55905, USA; Karpyak.Victor@mayo.edu

**Keywords:** prodynorphin, epigenetics, transcription, human brain

## Abstract

Neuropeptides serve as neurohormones and local paracrine regulators that control neural networks regulating behavior, endocrine system and sensorimotor functions. Their expression is characterized by exceptionally restricted profiles. Circuit-specific and adaptive expression of neuropeptide genes may be defined by transcriptional and epigenetic mechanisms controlled by cell type and subtype sequence-specific transcription factors, insulators and silencers. The opioid peptide dynorphins play a critical role in neurological and psychiatric disorders, pain processing and stress, while their mutations cause profound neurodegeneration in the human brain. In this review, we focus on the prodynorphin gene as a model for the in-depth epigenetic and transcriptional analysis of expression of the neuropeptide genes. Prodynorphin studies may provide a framework for analysis of mechanisms relevant for regulation of neuropeptide genes in normal and pathological human brain.

## 1. Introduction

Neuropeptides serve as neurohormones and local paracrine regulators. They control activity of neural circuits processing information relevant for behavioral, sensorimotor, endocrine and other processes [1]. Circuit functions and interaction between circuits are ultimately defined by cell-lineage and cell-type specific neuropeptide transcription that is regulated by epigenetic machinery.

Opioid peptides constitute the largest neuropeptide family. They include dynorphins, enkephalins, endorphins, and nociceptin/orphanin FQ that are processed from the prodynorphin (PDYN), proenkephalin, proopiomelanocortin, and pronociceptin precursor proteins. Effects of these peptides are mediated by κ- (KOR), δ-, μ-opioid receptors, and nociceptin receptor. Dynorphins are endogenous KOR ligands [2,3] acting through mitogen-activated protein kinases [4]. Dynorphins are expressed in the striatum, hippocampus, hypothalamus, amygdala and other brain areas, and, at the highest levels, in the pituitary gland suggesting their neuroendocrine functions [5,6].

The PDYN/KOR system regulates processing of reward, mood, nociception, stress response, and motor and cardiovascular functions [2,3,7,8,9,10]. Dysregulation of opioid peptides may cause depression, epilepsy and substance dependence [11,12,13,14,15,16,17]. Strikingly, mutations in dynorphins cause spinocerebellar ataxia SCA23 characterized by profound neurodegeneration in the brain of affected subjects [18,19,20]. Dynorphin expression and/or release is activated by stress [21]. The unique feature of KOR ligands is that they elicit dysphoric effects when administered to humans [22] and aversion in rodents [12,23,24]. Dysphoria and anxiety evoked by stress contribute to drug abuse in humans [25] and reinstatement of drug seeking in experimental animals [26]. Dysphoria induced by repeated stress is mediated by dynorphins [7]. The prodynorphin gene (*PDYN*) is identified as a hub associated with the neuroticism that predicts psychological disorders [27]. Experiments with KOR antagonists and gene deletion demonstrate that the endogenous dynorphins are involved in regulation of alcohol consumption and alcohol dependence [12,28,29,30,31]. Withdrawal developed due to discontinuation of drug use is severely dysphoric. The KOR mediated dynorphin effects may lead to negative mood and trigger “relief craving”, i.e., desire to suppress negative mood that often provokes drug-seeking in both human subjects and laboratory animals. Genetic studies associate polymorphisms in the *PDYN* gene and *OPRK1*, the KOR-encoding gene with heroin addiction, alcoholism, novelty seeking and positive reward traits [32,33,34,35,36,37]. *PDYN* variations are also linked to negative craving in alcohol-dependent subjects [38]. These pharmacological and genetic findings imply that KOR antagonists have a potential for treatment of depression and alcoholism including negative craving and relapse [39,40].

In this review, we focus on in-depth analysis of epigenetic and transcription mechanisms of *PDYN* regulation in the human brain. Understanding of these mechanisms could uncover general principles of regulation of neuropeptide genes that are specific for cell types, neuronal subtypes, and neural circuits. Selective regulation of neuropeptides by transcriptional and epigenetic mechanisms may underlie formation and rewiring of neural circuits in the human brain as the cellular basis of behavior and cognition.

## 2. Prodynorphin Transcripts and Proteins in the Human Brain

The human *PDYN* gene gives rise to mRNAs translated to the full-length (FL) and N-terminally truncated proteins (Figure 1 and Figure 2).

Two transcription start sites (TSSs) were identified in the human *PDYN* gene by targeted gene analysis [42] and FANTOM transcriptome analysis [43] (Figure 1c). The first TSS cluster determines the 5′-end of exon 1, while the second cluster is located in the coding segment of exon 4. *PDYN* mRNA giving rise to the full-length protein (FL1-*PDYN* mRNA) consists of four exons and three introns (Figure 2a). Testis-specific transcripts with alternative first exons (Taf I and Taf II) and the second FL mRNA (FL2), all differ from FL1 in exon 1 structure [42,44]. Three variants (GTEx 1-3) of dominant FL-PDYN transcripts that differ in the length of exon 1, the presence of exon 2 were identified by RNA-Seq analysis (http://www.gtexportal.org/home/, accessed on 6 May 2021 and Figure 2a). Sp1 and Sp2, and T1-T3 mRNAs are alternatively spliced 5′-truncated transcripts giving rise to N-terminally truncated (T) proteins (Figure 2b). A fragment of coding exon 4 is absent in Sp1, while exons 2 and 3 and a fragment of exon 4 are missed in Sp2 [42]. T1 and T2 are transcribed from the sites located between sequences coding for α-neoendorphin and dynorphin A. TSSs in exons are not unique for PDYN and were identified in other genes (Figure 2b) [45].

Exon 4 of human *PDYN* contains neuropeptide-encoding sequences, and also exhibits a promoter activity enabling transcription of T1- and T2-*PDYN* mRNAs from the intragenic TSSs. These variants give rise to N-terminally truncated 12 and 6 kDa PDYNs that lack a signal peptide. In cellular models, T1- and T2-proteins are targeted to the cell nucleus suggesting their non-canonical functions [42]. Transcripts initiated in exon 4 are produced in the amygdala, striatum and hippocampus (Figure 1c). The dynorphin encoding sequences are also hot spot for mutations that cause the neurodegenerative disorder spinocerebellar ataxia type 23 [18,19,20] (Figure 3a).

Besides FL1-*PDYN* mRNA, the gene gives rise to ∆SP*-PDYN* and ∆SP/NLS*-PDYN* mRNAs that are alternatively spliced variants (Figure 2b). ∆SP*-PDYN* mRNA contains two introns and three exons, and produces the ∆SP-PDYN protein that is translated from the Met_14_ and therefore lacks thirteen amino acids of the signal peptide (Figure 2b and Figure 3a). The ∆SP/NLS-PDYN protein in addition lacks α-neoendorphin and dynorphin A sequences [46]. ∆SP*-PDYN* mRNA is expressed in the striatum where its levels constitute approximately 30% of total *PDYN* mRNA, while in other brain areas its expression is negligible. ∆SP-PDYN protein has a putative bipartite nuclear localization signal (NLS) that has a high score and is located in the opioid domain [46]. The NLS targets ∆SP-PDYN protein to the cell nuclei. Biochemical methods and confocal imaging identified endogenous PDYN protein in the nucleus of neurons in the human striatum (Figure 3b–d) [46]. Consistently, electron microscopic analysis of rat nucleus accumbens demonstrated the presence of Pdyn protein and dynorphin A in the neuronal nuclei along with its location in the smooth endoplasmic reticulum [47]. Proenkephalin, another opioid peptide precursor, was found in the cell nucleus in several cell lines [48,49].

Bioinformatics analysis of PDYN, proenkephalin and proopiomelanocortin predicts that these opioid peptide precursors may serve as DNA-binding proteins. They have zinc-finger and helix-loop-helix domains that are similar to those of twist, hunchback, tal and lil-1 transcription factors [50]. The cystein-rich pattern is perfectly conserved in the opioid peptide precursors and fits to the pattern of zinc-finger domains of transcription factors. Furthermore, the enkephalin sequences represent heptapeptide repeats typical for helix-loop-helix DNA-binding motif.

Nuclear localization of neuropeptide precursor proteins is unusual phenomenon that along with structure similarity with transcription factors predicts a novel transcriptional and/or epigenetic function of these proteins. This function may be essential, at least for *PDYN*, for area-specific regulations. This is supported by the presence of ∆SP-*PDYN* mRNA in the human striatum but not in other human brain regions.

Long non-coding RNAs are RNA molecules that are not translated into proteins and may function as gene-specific regulators of transcription and epigenetic modifications. The *AK090681* gene is transcribed from the opposite strand relative to *PDYN* in the locus and gives rise to non-coding RNA [51]. The nucleus accumbens and cerebellum strongly differ in *PDYN* and *AK090681* expression. The levels of *PDYN* mRNA are 1,000-fold higher while those of the *AK090681* RNA are 20-fold lower in the nucleus accumbens vs. the cerebellum. Long non-coding *AK090681* RNA may be involved in regulation of *PDYN* transcription (see Section 5.1).

In conclusion, transcription of the human *PDYN* gene is highly plastic resulting in generation of a variety of mRNAs that give rise to several proteins serving as opioid peptide precursors, or nuclear proteins that may regulate transcriptional and epigenetic processes.

## 3. *PDYN* Promoter Mapping and Identification of Transcription Factors

The conservation of *PDYN* promoter is weak across vertebrates besides short, about 300 nucleotides segment located upstream of the main TSSs (Figure 1b). At the same time, the 1.25 kb *PDYN* promoter region shows similarity among humans, great apes and monkey (Figure 1b). This region may be an example of the recent unique positive selection of cis-regulation in human genome [52]. The low similarity between human and rodent suggests that rodent models are not suitable for analysis of *PDYN* regulation associated with human disorders.

The conserved 300 bp of human *PDYN* promoter fragment is responsible for basal and protein kinase A-activated transcription. This fragment includes a downstream response element (DRE) that mediates transcriptional repression. DRE is a DNA binding element for the transcriptional repressor downstream regulatory element antagonist modulator (DREAM), the Ca^2+^-binding transcriptional repressor [53]. DREAM inhibits *PDYN* transcription while its genetic deletion results in upregulation of expression of this gene [54]. Complexing with alphaCREM, the CREM repressor isoform prevents DREAM–DRE binding and allows the cyclic AMP-dependent *PDYN* de-repression [55,56].

NF-κB, a nuclear factor kappa-light chain-enhancer of activated B cells, and YY1, Yin-Yang1 transcription factors may regulate *PDYN* transcription [57,58,59]. They target the exon 4 *PDYN* DNA sequences that encode dynorphin peptides. These findings suggest that the neuropeptide sequences that are short, well conserved and present in several copies in neuropeptide-encoding genes, may serve as binding elements for sequence-specific transcription factors. These unique sequences may represent DNA signatures—identifiers of the neuropeptide genes, allowing their selective transcriptional regulation. This hypothesis is supported by the findings that the neuropeptide encoding sequences in exon *PDYN* 4 may function as gene promoter and activate transcription of a reporter gene [42].

Animal studies suggest that *Pdyn* may be regulated by ΔFosB, a component of Activator protein 1 (AP-1) transcription factor that consists of two protein subunits [60]. ΔFosB, a truncated FosB protein was proposed as the major transcriptional integrator in addictive, stress and psychiatric disorders [61]. Still, no detailed transcriptional analysis of ΔFOSB in human brain supports this hypothesis, and no study assessed yet if ΔFOSB is present in the AP-1 complex and regulates human *PDYN* transcription. The expression levels of ΔFOSB are very low or negligible in the human brain compared to those of two other FOSB proteins. No changes in the levels of this protein were detected in addicted human brain, and this protein is not present in the AP-1 transcription factor that targets the *PDYN* AP-1 binding element [62]. Instead, the AP-1 dimer consists of FOSB and JUND subunits in human brain. Thus, there is no evidence that ΔFosB is involved in human addiction disorders [62].

## 4. Genetic Factors Contributing to *PDYN* Regulation

Human studies identified strong associations of SNPs in *PDYN* with alcoholism, drug addiction, emotions and memory. Alcoholism and alcohol dependence are associated with several SNPs in the *PDYN* 5′-promoter, exons 3 and 4, and 3′-untranslated region (3′-UTR) [35]. The 3′-UTR contains of six SNPs that form a haplotype block associated with alcohol dependence [35]. The risk haplotype is also associated with combined cocaine dependence and cocaine–alcohol co-dependence along with cocaine dependence, and likely enables low *PDYN* expression in the caudate and nucleus accumbens [63]. The *PDYN* rs2281285-rs1997794 haplotype is associated with alcoholism and susceptibility for drinking in negative emotional states [38,64].

Variable number tandem repeats (VNTR) are often associated with complex disease traits. The 68 bp VNTR is located upstream of the *PDYN* main TSSs (Figure 4b). The VNTR copy number varies from one to five in humans while a single copy is present in nonhuman primates, and none in other animals [52]. The human VNTR elements have five substitutions that differentiate them from chimpanzees. The DNA sequence similar to the AP-1 binding element is located in the *PDYN* VNTR and may serve as a target for this transcription factor [65]. The VNTR elements may contribute to *PDYN* regulation that is dependent on their number, and cellular context [66]. The *PDYN* VNTR variants are associated with epilepsy [11,67], cocaine dependence and abuse [68], schizophrenia [69], opioid addiction [38,70,71], methamphetamine dependence [72,73], and cocaine/alcohol co-dependence [74]. However, attempts to replicate these studies were generally unsuccessful [6,75,76].

## 5. *PDYN* Epigenetic Mechanisms

### 5.1. PDYN Regulation in Chromosomal Context

Strong signals of CTCF, the CCCTC-binding factor in the *PDYN* locus were detected in a variety of cell lines in the genome-wide screen (Figure 4a). CTCF is a pleiotropic transcription factor that may activate or repress gene transcription, and contribute to gene insulation and imprinting. CTCF possesses eleven zinc fingers that may bind to diverse DNA sequences and, by this virtue, may mediate intra- and interchromosomal interactions by chromatin looping between insulators targeted by CTCF [77]. Chromatin domains in these loops may be either activated [78] or repressed for their gene transcription [77] through facilitation or inhibition of interactions of the enhancers and inhibitors with gene promoters. Strong peaks of CTCF are located at putative boundaries of the *PDYN* gene (Figure 4a) where they overlap with strong REST, MYC, MAX, USF1, EGR1 and ZNF143 signals. Localization of the sites occupied by CTCT corresponds to exons 3 and 4 of long non-coding *AK090681* RNA that are positioned on the complementary DNA strand (Figure 4a).

Besides CTCF, the *PDYN* locus is regulated by RE1-Silencing Transcription Factor (REST) that targets the neuron-restrictive silencer element RE-1 and acts as a transcriptional repressor (Figure 4a; see Section 5.4). REST forms two peaks with the first located upstream of the gene in proximity to the upstream CTCF site proximally to the gene. The second peak is in the 3′-UTR. The emerging mechanism is that these two transcription factors may control the locus specific *PDYN* transcription that is insulated from *AK090681* by formation of the chromatin loops due to CTCF binding (Figure 4a). Similar to other long non-coding RNAs, the *AK090681* long non-coding RNA may be involved in the regulation of gene transcription by coordinating intrachromosomal looping and recruiting the chromatin modifying factors.

Lack of cellular and animal models is a limiting factor in analysis of human *PDYN* transcription in chromosomal context. Cell lines of rodent and human origin do not generally express prodynorphin or transcribe it at much, approximately 1000-fold lower levels compared to the brain. Furthermore, in in vitro cellular models *PDYN* transcription is not responsive to pharmacological treatments that upregulate expression of this gene in animal models.

### 5.2. DNA Methylation

The *PDYN* gene is transcribed mostly in neurons in the human brain [79,80]. The neuronal expression may be controlled through methylation of two short adjacent differentially methylated regions, DMR1 and DMR2 in the promoter. Methylation patterns are opposite between neurons and glia for each DMR, and also between the DMRs (Figure 5). DMR1 comprises of a short, nucleosome size CpG island (CGI) that is hypomethylated and enriched in 5-hydroxymethylcytosine in neurons, while hypermethylated in other cell types.

The current paradigm is that DMRs are associated with CGI shores but not with core of the CGIs [81,82]. In contrast to this pattern, methylation of the *PDYN* CGI is different among neurons and other cell types. This CGI is also enriched in 5-hydroxymethylcytosine suggesting its function as active regulatory domain characterized by high cytosine methylation turnover rate [79]. Methylation of individual CpGs in the CGI is highly coordinated in neurons, that is not observed in other *PDYN* promoter areas in these and other cell types. This pattern in the *PDYN* CGI is analogous to contiguous methylation clusters characterized by high correlations among CpG methylation in other genes [83].

The CGI methylation may differ in its chromatin organization between neurons and glia. In cells that do not transcribe the *PDYN* gene, the CGI is wrapped in a nucleosome, a feature of the repressive chromatin [79]. Thus, the CGI may serve as the *PDYN* promoter module, which cycling between the methylated and occupied by a nucleosome state, and hydroxymethylated nucleosome free state, is locally regulated. These DNA and chromatin modifications may allow interactions with sequence–specific transcription factors that could delineate cell-type specific *PDYN* transcription.

The opposite pattern is observed for DMR2 that is hypermethylated in neurons and hypomethylated in glia and other cell types (Figure 5). This pattern and also negative correlations between methylation of the two DMRs imply that the mechanisms that are autonomic for each DMR and coordinated between them, regulate their methylation, and consequently their complementary role in *PDYN* expression. Hypermethylation of DMR2 in neurons may enable binding of MeCP2 or other factors that binds to methylated DNA and activate gene transcription. In non-neuronal cells, *PDYN* may be repressed through interactions of DREAM, a methylation sensitive transcriptional repressor, with non-methylated DMR2 [84]. There epigenetic mechanism may control cell type-specific control of *PDYN* transcription in human brain.

One more CpG island is located in the coding part of exon 4 of the *PDYN* gene (Figure 4b). High methylation of cytosine residues in exons [85] were detected in many genes that was positively associated with gene transcription levels [86,87]. The exon 4 *PDYN* CpG island possesses a promoter activity, transcription factor binding sites, and the second cluster of TSSs, along with SNP associated with alcoholism [35,42,57,58,59]. This island is similarly hypermethylated in the brain and blood DNA in which the levels of FL-*PDYN* transcript are high or negligible, respectively [80]. DNA methylation profile in this domain is well conserved across human individuals, whereas differs among brain and peripheral tissues, and among brain regions [79]. Epigenetic mechanisms may remodel chromatin structure in the exon 4 *PDYN* CpG island that may result in transcription of this gene from intragenic TSSs or regulate elongation of transcription and mRNA splicing.

### 5.3. The CpG-SNP Hypothesis: Epialleles of PDYN SNPs Associated with Alcoholism

The current paradigm is that the environmental, epigenetic and genetic factors influence the phenotype and contribute to the propensity for disorders by altering gene transcription. SNPs are much more abundant at the CpG dinucleotides than predicted [88] and may form or disrupt a CpG sequence. Methylation and hydroxymethylation of CpG-SNPs is allele-specific. Environmental, epigenetic and genetic factors may mechanistically integrate on CpG-SNPs which genetic variants may determine the phenotype while the cytosine methylation—the demethylation state may control transcription from the C-allele (Figure 6a). Two SNP alleles and three cytosine epialleles including its unmethylated, methylated and hydroxymethylated states may differentially contribute to a vulnerability of a disease.

Five *PDYN* SNPs are associated with alcoholism with high significance [89]. Three of them form or disrupt CpG sites (rs1997794; rs6045819 and rs2235749; Figure 6b). To test the CpG-SNP hypothesis we analyzed methylation of these three *PDYN* CpG-SNPs in the human dlPFC. Alcoholism is associated with hypermethylation of the C allele of 3′-UTR CpG-SNP rs2235749 (C > T) in the human brain, and its methylation levels positively correlate with *PDYN* expression suggesting a functional link between these two processes. Analysis of DNA-binding factors targeting this area identified a novel T-allele-binding factor (Ta-BF). This 63 kDa protein has high affinity for the T and methylated C alleles of the 3′-UTR CpG-SNP but not for unmethylated C allele (Figure 6c).

Positive correlation between the 3′-UTR CpG-SNP methylation and *PDYN* expression suggests that the Ta-BF binding to the 3′-UTR may activate *PDYN* transcription [89]. Thus, the environmental, epigenetic and genetic factors associated with alcoholism may be mechanistically integrated on the *PDYN* 3′-UTR CpG-SNP, and Ta-BF may read the resulting methylation signals and translate them into disease predisposition through changes in *PDYN* transcription.

Several gene-centric and genome wide human studies lend support for the CpG-SNP hypothesis. Many well-known polymorphic sites associated with psychiatric disorders form CpG-SNPs. CpG-SNPs of the catechol-O-methyltransferase, GABA(A) receptor beta(2) (*GABRB2*) and μ-opioid receptor genes are the examples. Methylation of the cytosine allele at these sites is a part of the mechanism that controls gene transcription and contributes to the phenotype [90,91,92,93,94,95]. Thus, modifications of CpG-SNPs may have an essential epigenetic function that mediates the effects of a changing environment on the polymorphism dependent genome expression.

### 5.4. PDYN Regulation by REST

REST is a master regulator of neuronal phenotype acting through neuron-restrictive silencer element (RE1) and inhibiting transcription of its target genes [96,97]. The REST effects are mediated by epigenetic mechanisms that recruit inhibitory enzymatic activities to its target elements leading to long-term alterations in gene transcription.

Chromatin immunoprecipitation data generated by ENCODE [98,99] demonstrate that *PDYN* has two binding sites for REST (Figure 4a) [100,101]. They are located 12 kb upstream of the *PDYN* gene and in its 3′-UTR, respectively [102]. Functional inactivation of REST with a dominant negative mutant REST protein [103,104] increases *PDYN* transcription in cellular models [104]. Consistently analysis of the human dlPFC by Chromatin Immunoprecipitation quantitative real-time PCR assay revealed REST bound to the RE1 located upstream of *PDYN*, while binding to the 3′-UTR RE1 element was negligible [104].

REST is regulated by the microRNA MIR-9, and they together control chromatin remodeling that determines cell phenotype [105,106]. Analysis of the human brain demonstrates that REST and MIR-9 negatively correlate suggesting the negative feedback mechanism. Thus, REST may repress *PDYN* transcription while this transcription factor is negatively controlled by MIR-9 microRNA [104].

### 5.5. Dual Epigenetic and Transcriptional Mechanism Controls Neuronal PDYN Expression

Expression of the neuropeptide genes including *PDYN* is confined to specific cell types and neuronal lineages that may be coordinated by epigenetic and transcriptional mechanisms. These mechanisms may permit and restrict, activate or inhibit gene transcription depending on cell type. We tested this hypothesis by the in-depth analysis of the opioid *PDYN* transcription in the human brain [79]. Our strategy was to detect sequences in the *PDYN* locus that are differentially methylated between neurons and other cell types, and to identify sequence-specific methylation-sensitive transcription factors that target these DMRs, and therefore may control the methylation-regulated *PDYN* expression in specific cell types.

The previous sections described that the *PDYN* promoter has the DMR1 with a short CGI as a core that is hypomethylated and enriched in 5-hydroxymethylcytosine in neurons (Figure 5) [79]. When unmethylated this CGI serves as a binding site for USF2, E-box transcription factor that does not interact with methylated sequences. USF2 activates *PDYN* transcription in model cell systems, and is physically associated with unmethylated E-box in the *PDYN* CGI in human brain. Consistently, expression of USF2 and *PDYN* is correlated (Figure 7a,b). USF2 and PDYN proteins are co-expressed in the same neurons in the human dlPFC; only USF2-producing cells synthesize dynorphins (Figure 7c–e). Thus, two conditions may be obligatory for the neuron-specific *PDYN* transcription that are the CGI hypomethylation and USF2 expression.

In rodents, Ptf1a, Pax2, Neurod1/2/6 and Bhlhb5 transcription factors enable *Pdyn* expression in cell type and cell lineage-specific patterns in the dorsal spinal cord and Islet-1 in the striatum [107,108,109,110,111,112]. USF2 and these transcription factors are E-box proteins (USF2, Ptf1a and Neurod1/2/6), or regulate E-box dependent transcription. Thus, formation of cellular prodynorphin phenotype in the human and rodent central nervous system is determined by E-box transcription factors.

## 6. *PDYN* Transcriptional Adaptations Concomitant with Neuronal Decline in Human Alcoholics

Alcoholism is associated with cognitive impairments that may develop due to aberrant neurotransmission and neurodegeneration. Several lines of evidence suggest that dynorphin opioid peptides have a role in cognitive decline [113,114,115,116,117]. In animal experiments, dynorphins administered into the hippocampus impair spatial learning [118]. Dynorphins also contribute to the stress and age-related learning and memory deficits [113,114,115]. In elderly humans, *PDYN* polymorphism is associated with memory loss [116]. Dynorphins are elevated in the prefrontal cortex of patients with Alzheimer’s disease, and their levels correlate with neuropathological score [117]. Consistently, it was hypothesized that the PDYN/KOR system is dysregulated in the dlPFC and hippocampus of alcoholics, and that these changes contribute to cognitive impairments [119,120]. This notion gains a support in animal model of cognitive deficits induced by alcohol binge drinking [120,121,122]. The PDYN/KOR system mediated impairments of spatial learning and memory in this model while selective κ-antagonist nor-binaltorphimine reversed these impairments by decrease in the ethanol-induced elevation of glutamate overflow.

In line with these studies, changes in the PDYN/KOR system are considered as a molecular mechanism that underlies the long-term effects of addicted substances on behavior, cognitive impairment and loss of control over intake of addictive substances and alcohol [12,30,120,123,124,125,126,127,128,129]. This hypothesis was addressed by analysis of the expression levels and co-expression (transcriptionally coordinated) patterns of *PDYN* and *OPRK1* (KOR) genes in the dlPFC of alcoholics; 53 alcoholics were compared with 55 control subjects [127,128]. *PDYN* was found to be downregulated in the addicted brain, while the *OPRK1* expression was not altered. Thus, the effects of alcoholism on these two genes were not mechanistically coordinated.

Early postmortem morphological studies revealed marked reduction in the number of neurons in the dlPFC of alcoholics [130,131]. More recently this was confirmed by analysis of neuronal proportion in the epigenome-wide DNA methylation study, and by analysis of neuronal and glial markers [132]. Importantly, the *PDYN* mRNA levels were not affected by the decline in the neuronal number. Instead, their alterations were likely caused by transcriptional adaptations [79,127,128]. Another issue that is important for regulation of the PDYN/KOR system is that the absolute levels of *PDYN* mRNA were markedly lower compared to those of KOR (*OPRK1*). Thus, *PDYN* transcription may be a limiting factor in the PDYN/KOR signaling. Therefore, a decrease in *PDYN* transcription may further diminish efficacy of PDYN/KOR signaling in dlPFC of alcoholics. Resulting overactivation of neurotransmission in cortical neurocircuits that is negatively controlled by dynorphins, may be a gross effect of PDYN/KOR downregulation that contributes to formation of alcohol-induced impairments in behavior.

Addictive substances may cause similar downstream molecular adaptations—the general molecular syndrome that mediates the lasting nature of the addictive state [133]. *PDYN* transcription is downregulated in the dlPFC and dorsal striatum in alcoholics [127,128,129], and in the dorsal striatum in cocaine addicts [63]. These changes may be a part of the general adaptive syndrome caused by addictive substances.

## 7. Conclusions

The specific feature of neurons and neural circuits is their neuropeptide phenotype. In short, conserved and repetitive neuropeptide sequences are a distinctive feature of neuropeptide genes. In this property these sequences are similar to DNA-binding sites for sequence-specific transcription factors. The opioid peptide sequences in the *PDYN* gene overlap with or are situated in close proximity to multiple TSSs, splice junctions, and CpG-SNP that is associated with psychiatric phenotype (Figure 6b). YY1 and NF-κB through binding to these sequences may activate transcription from a cryptic promoter located in this area [42,57,58,59]. This region is hypersensitive to DNAase I, suggesting that chromatin is open in the short CpG island that is a core of this region (Figure 4b). Strikingly, the dynorphin-encoding sequences may be a mutational hot spot; their missense mutations cause profound neurodegeneration in human subjects with neurodegenerative disorder SCA23 [20,134,135,136]. SCA23 mutations are enriched in CpGs suggesting a link of their origin or a pathogenic mechanism with methylation/demethylation processes [79,137]. Together these findings lend support for the hypothesis that the neuropeptide-encoding sequences may have regulatory functions. They may have a role in transcriptional initiation, elongation, and RNA splicing leading to synthesis of transcripts that give rise to protein variants with non-canonical functions. A unique combination of transcriptional mechanisms regulated by conventional enhancer and promoter, and by the neuropeptide-encoding sequences may determine cell-type and cell lineage-specific gene expression. Whether neuropeptide-encoding sequences are gene signatures targeted by epigenetic mechanisms that define neuropeptide phenotype of neural circuits, is important to address in future studies.

Dysregulation of neural circuits expressing neuropeptides may cause neurological and psychiatric disorders including spinocerebellar ataxia 23, epilepsy, depression and substance dependence. Not all functions of neuropeptide precursors might have been identified. The unusual nuclear localization of PDYN and proenkephalin was demonstrated, and predicts novel epigenetic or transcriptional functions for nuclear variants of these proteins. Knowledge of the mechanisms that regulate epigenome and transcriptome in the neuropeptide producing neurons is essential for understanding of normal and dysfunctional neural circuits. Studies focused on individual neuropeptide genes and functions of their protein products complement, specify and further advance multi-omics analysis of human brain relevant for psychiatric disorders.

## Figures and Tables

**Figure 1 molecules-26-03458-f001:**
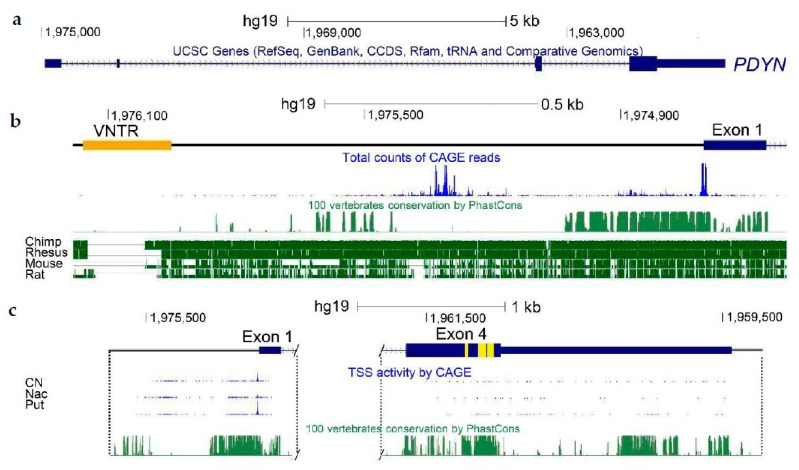
Human *PDYN* gene (modified screenshot from UCSC Genome Browser). (**a**) Gene structure. (**b**) Promoter *PDYN* region with VNTR and TSSs. Conservation across vertebrates. (**c**) Canonical *PDYN* mRNAs and transcripts initiated in exon 4. Their conservation across vertebrates. Non-coding sequences are shown by thin *dark blue* line; coding sequences by thick *dark blue* line; dynorphin peptides-encoding sequences by *yellow*. CN, caudate nucleus; NAc, nucleus accumbens; Put, putamen. Modified from [41].

**Figure 2 molecules-26-03458-f002:**
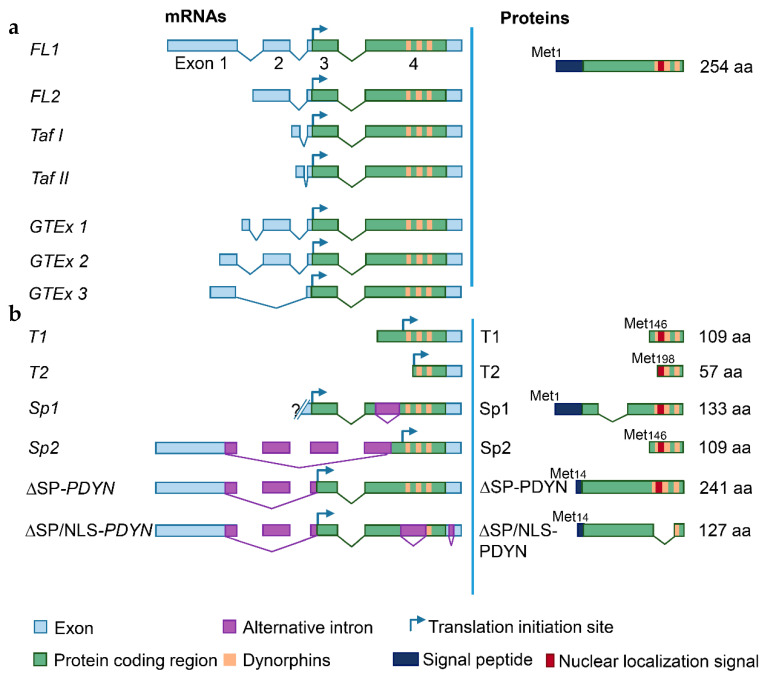
*PDYN* mRNAs coding for the full-length (FL, (**a**)) and truncated proteins (**b**). (**a**) Transcripts encoding FL-PDYN protein. The dominant FL1-*PDYN* and shorter transcripts including FL2-*PDYN* and *GTEx1-3* and testis-specific *Taf* I and *Taf* II transcripts differ in the first and second exons, and in TSS. (**b**) PDYN mRNAs encoding truncated PDYN proteins including alternatively spliced Sp1, Sp2, ΔSP-PDYN and ΔSP/NLS-PDYN transcripts, and transcripts initiated within the coding part of exon 4 (T1 and T2). Signal peptide is truncated in both ΔSP- and ΔSP/NLS-PDYN. Putative nuclear localization signal (NLS) is located in the dynorphin domain. Curved arrows show initiation of translation. Modified from [46].

**Figure 3 molecules-26-03458-f003:**
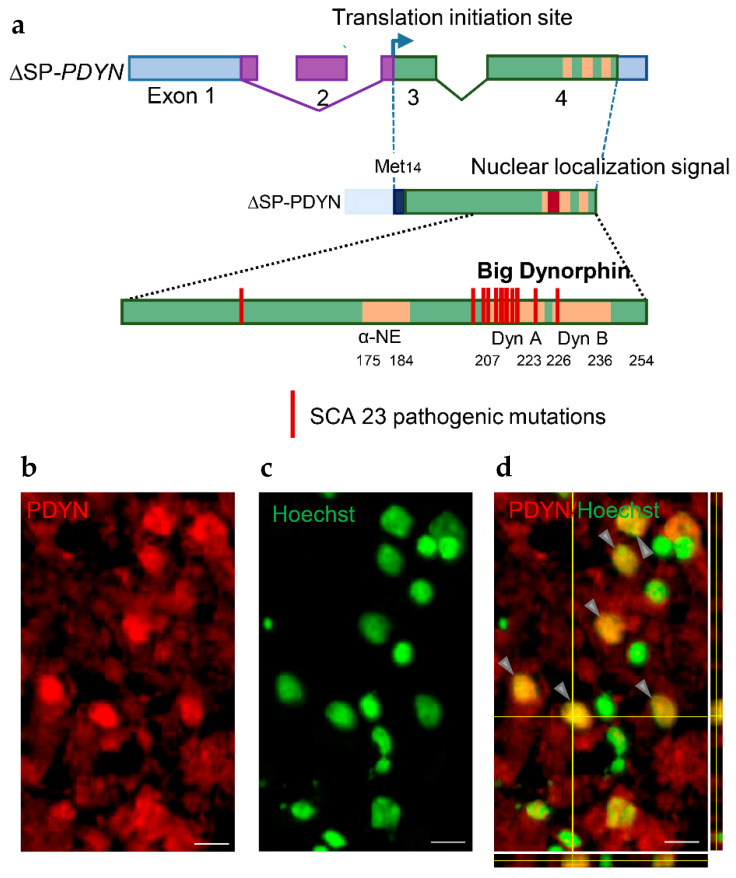
Structure of ΔSP-*PDYN* mRNA and protein, *PDYN* pathogenic mutations causing SCA23, and nuclear localization of ∆SP-PDYN protein. (**a**) ΔSP*-PDYN* encode ΔSP-PDYN protein with truncated signal peptide. Sequences of opioid peptides α-neoendorphin (α-NE), dynorphin A (Dyn A), dynorphinB (Dyn B), and big dynorphin (Big Dyn) are shown in yellow. Pathogenic mutations form a mutational hot spot that is localized within the pathogenic big dynorphin sequence with dynorphin A as a core. (**b**,**c**) PDYN immunoreactivity (red) in the nuclei (green) of neurons in the human caudate nucleus. (**d**) Double labeling (yellow) of neuronal nuclei (arrows) in 3D confocal reconstruction projections. Scale bar, 20 μm. Modified from [46].

**Figure 4 molecules-26-03458-f004:**
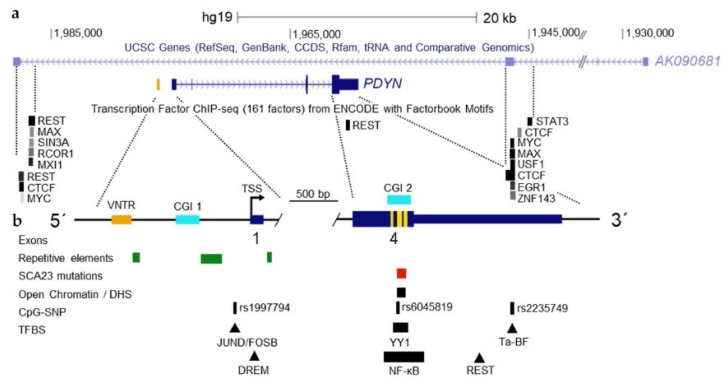
Locus of human *PDYN* with targets for transcription factors. (**a**) Genomic organization showing *PDYN*, the antisense *AK090681* transcript and transcription factor targets deposited on UCSC Genome Browser. (**b**) Verified and putative transcription factor binding elements, promoter VNTR, CpG islands 1 and 2 (CGI 1 and CGI 2), *PDYN* pathogenic mutations causing neurodegeneration, and DNase I hypersensitivity sequence (DHS), and CpG-SNPs association with alcoholism. Thin *light blue* line shows non-coding RNA, thick *dark blue* line coding region, vertical *yellow* lines dynorphin sequences. Modified from [41].

**Figure 5 molecules-26-03458-f005:**
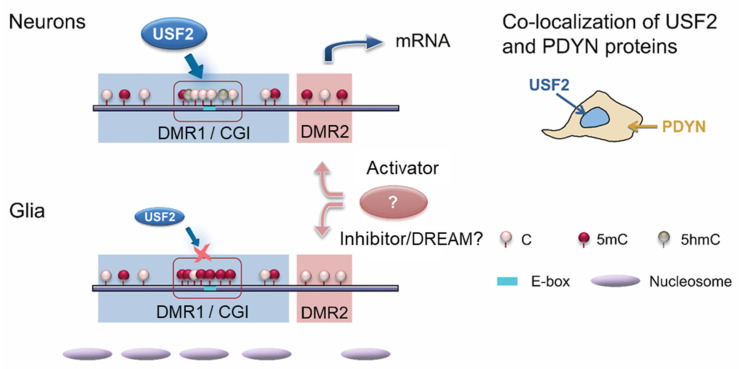
Model for epigenetic and transcriptional regulation of neuronal *PDYN* transcription. In neurons, USF2 binds to E-box in the promoter CGI that is hypomethylated and enriched in 5-hydroxymethylcytosine (5-hmC). In glia, the CGI is hypermethylated. DMR2 and DMR1/CGI exhibit methylation patterns that are opposite between them and between neurons and glia for each of them. In non-neuronal cells, DMR2 may be targeted by methylation-sensitive transcriptional repressor such as DREAM, while in neurons by a methylation-dependent transcriptional activator. In glia, the DMR1/CGI may be wrapped in a nucleosome, that prevents transcriptional initiation. These mechanisms may underlie contrasting *PDYN* expression in neurons and glia. Modified from [79].

**Figure 6 molecules-26-03458-f006:**
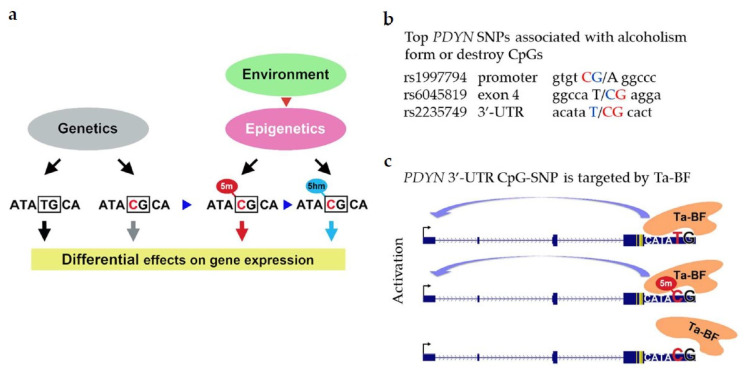
The CpG-SNP hypothesis. (**a**) Genetic, epigenetic and environmental factors are mechanistically integrated at CpG-SNPs that may be methylated and hydroxymethylated at the C-allele. Two alleles and three cytosine epialleles may differentially affect gene transcription and thereby differently contribute to deasease predisposition [89]. (**b**) *PDYN* SNPs variants associated with alcoholism are shown in *blue* while those forming CpGs in *red*. (**c**) T-allele-binding factor (Ta-BF) has high affinity for the T and methylated C alleles of the 3′-UTR CpG-SNP but not to unmethylated C allele. The high affinity interaction may be a basis for transcriptional activation by this DNA-binding protein.

**Figure 7 molecules-26-03458-f007:**
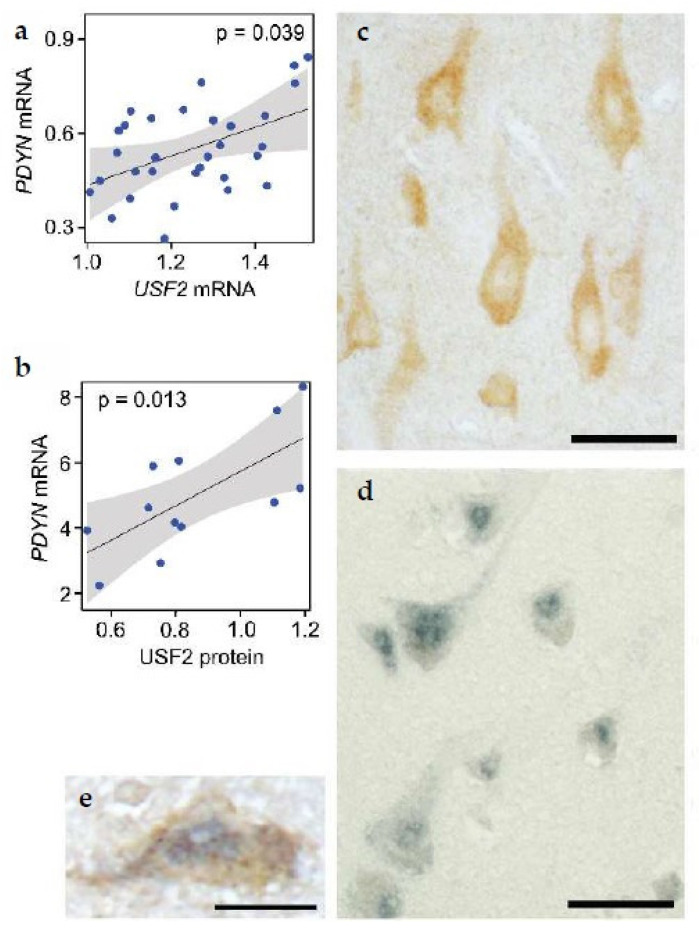
Correlation of USF2 and PDYN (**a**,**b**), and their co-localization (**c**–**e**) in the human dlPFC. (**a**,**b**) The estimated effect with 95% confidence interval. Immunoreactivity of (**c**) PDYN, and (**d**) USF2 in the cytoplasm and nuclei of the layer V neurons, respectively. (**e**) Double labeling of PDYN and USF2 in the same neuron. Scale bars, 50 μm (**c**,**d**); and 25 μm (**e**). Modified from [79].

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
