# Peer review of "Epigenetic and Transcriptional Control of the Opioid Prodynorphine Gene: In-Depth Analysis in the Human Brain"

_molecules, 2021, doi:10.3390/molecules26113458_

Round 1
Reviewer 1 Report
The manuscript “epigenetic and transcriptional control of opioid gene: in-depth analysisi in the human brain” by Olga Nosova et al. is a very comprehensive collection of presently available information. It is well written and addresses all important aspects.
There are only a few comments from my side:
I would very much appreciate a chapter on how these regulatory mechanisms may act in patho-physiologies involving the dynorphin / kappa opioid receptor system. So far the manuscript reflects on pre-disposition only, but not on how the expression of PDYN is regulated post- and interictally in epilepsy for example. Or how the balance of MOR and KOR signaling is shifted in addiction.
In the legend to figure 7, line 453, authors state that “both proteins have predominantly nuclear localization”. This is not supported by the labeling for PDYN in c. Labeling is moderate in the cytoplasm and very low in nucleus.
Minor points:
There are some typos like two .. after ref 38 in line 56.
In line 100 “PDYN gene dives rise” probably should read PDYN gene gives rise”.
In line 167 “therfore” should read “therefore”
dlPFC is inconsistenly termed dl-PFC in line 410
The short paragraph from line 114 to 118 doesn’t really fit here. Maybe it can be omitted totally.
Author Response
We thank you for analysis of our manuscript and advices. We agree with the comments and suggestions, and have addressed them to the best of our ability. The manuscript has been substantially modified and cleaned up, and a new section requested by the first reviewer has been added.
- “I would very much appreciate a chapter on how these regulatory mechanisms may act in patho-physiologies involving the dynorphin / kappa opioid receptor system. So far, the manuscript reflects on pre-disposition only, but not on how the expression of PDYN is regulated post- and interictally in epilepsy for example. Or how the balance of MOR and KOR signaling is shifted in addiction.”
RESPONSE: We thank the reviewer for suggesting this option. A new section (Section 6) focusing on the transcriptional dysregulation of the PDYN/KOR system in addicted human brain has been added to the manuscript. Virtually no other reports with in-depth mechanistic transcriptional and epigenetic analysis of gene regulation in pathological human brain are available at the moment, or they explore a very limited number of subjects, that undermines their significance and therefore they have not been discussed in the review.
- “In the legend to figure 7, line 453, authors state that “both proteins have predominantly nuclear localization”. This is not supported by the labeling for PDYN in c. Labeling is moderate in the cytoplasm and very low in nucleus.”
RESPONSE: We are sorry about this mistake. It has been corrected. PDYN is located in the cytoplasm while USF2 predominantly in the nucleus of neurons. Images were replaced by the better ones.
- “Minor points”
RESPONSE: Typos have been corrected and other minor remarks have been addressed accordingly.
Reviewer 2 Report
The review deals with epigenetic and transcriptional control of dynorphin gene in human brain.
The review is interesting and clear enough to me, even if there are some points I would like to raise for the sake of clarity.
- First of all the review deals only with dynorphin/KOR system whereas the title contain the generic term opioid. I would suggest to change the title indicating dynorphin system.
- In the Introduction the Authors state that the opioid system include 3 endogenous families, without citing the fourth member orphaninFQ/NOP family. I suggest to include some information on that one, too.
- In Introduction, on page 2, line 52, something wrong happened, in fact it is not correct that dynorphin may lead to relief craving, since it is well known that KOR antagonists may be able to treat relapse. I think that the Authors need to check is some other words are lacking and I suggest to rephrase correctly the sentence in order to be clear.
- There are many typing mistakes that affect the potential interest for the review.
- Add more information in figure legends to indicate abbreviations, for example in figure 2 what T1 or T2 stand for.
- On page 4, line 169 what rich stand for? You may wanted to say: reach about the 30%?
- On page 6, line 234 take away of!
- On page 10, line 411: what RESF stand for? is maybe REST
- I also would suggest to the Authors to take care of the Abbreviations throughout the review for increasing its readibility.
Author Response
We thank you for analysis of our manuscript and advices. We agree with the comments and suggestions, and have addressed them to the best of our ability. The manuscript has been substantially modified and cleaned up.
- “First of all the review deals only with dynorphin/KOR system whereas the title contain the generic term opioid. I would suggest to change the title indicating dynorphin system.
In the Introduction the Authors state that the opioid system include 3 endogenous families, without citing the fourth member orphaninFQ/NOP family. I suggest to include some information on that one, too.
In Introduction, on page 2, line 52, something wrong happened, in fact it is not correct that dynorphin may lead to relief craving, since it is well known that KOR antagonists may be able to treat relapse. I think that the Authors need to check is some other words are lacking and I suggest to rephrase correctly the sentence in order to be clear.
There are many typing mistakes that affect the potential interest for the review.
Add more information in figure legends to indicate abbreviations, for example in figure 2 what T1 or T2 stand for.
On page 4, line 169 what rich stand for? You may wanted to say: reach about the 30%?
On page 6, line 234 take away of!
On page 10, line 411: what RESF stand for? is maybe REST
I also would suggest to the Authors to take care of the Abbreviations throughout the review for increasing its readibility..”
RESPONSE: We thank the reviewer for the detailed analysis of the manuscript. All comments have been addressed and necessary correction made.
Round 2
Reviewer 2 Report
I think that now the manuscript is improved and accetable for publication.
All questions have been detailed answered by the Authors